# Characteristic of Oral Squamous Cell Carcinoma Tissues Using Isotope Ratio Mass Spectrometry

**DOI:** 10.3390/jcm9113760

**Published:** 2020-11-22

**Authors:** Katarzyna Bogusiak, Aleksandra Puch, Radosław Mostowski, Marcin Kozakiewicz, Piotr Paneth, Józef Kobos

**Affiliations:** 1Department of Maxillofacial Surgery, Medical University of Lodz, 1 Gen. J. Hallera Pl., 90-647 Lodz, Poland; puch.aleksandra@wp.pl (A.P.); marcin.kozakiewicz@umed.lodz.pl (M.K.); 2Institute of Food Technology and Analysis, Faculty of Biotechnology and Food Sciences, Lodz University of Technology, 4/10 Stefanowskiego street, 90-924 Lodz, Poland; radoslaw.mostowski@p.lodz.pl; 3Institute of Applied Radiation Chemistry, Lodz University of Technology, 116 Żeromskiego, 90-924 Lodz, Poland; piotr.paneth@p.lodz.pl; 4Department of Histology and Embriology, Medical University of Lodz, 7/9 Żeligowskiego Street, 90-752 Łódź, Poland; jozef.kobos@umed.lodz.pl

**Keywords:** oral squamous cell carcinoma, tumour, stable isotopes, IRMS, spectrometry, isotopic analysis

## Abstract

Overall prognosis for patients with oral squamous cell carcinomas (OSSC) is still unfavourable. However, there is a hope that a novel diagnostic method may establish better cancer biology characteristics. The aim of this study was to evaluate the isotope ratio of nitrogen and carbon in OSSC as compared to margin and healthy tissue. A total of 18 patients with OSSC were included in the study. Specimens collected covered: four tumour, four margin and two healthy oral mucosa samples. The samples underwent further procedures: lyophilization and isotope ratio mass spectrometry. Measurements of the ratio of stable isotopes of nitrogen ^15^N/^14^N and carbon ^13^C/^12^C were performed. It is noticeable that the highest average nitrogen concentration was observed in tumour 12 ± 0.4% and the lowest in healthy tissues 8 ± 0.9% (*p* < 0.00001). The highest average carbon content was observed in healthy tissues 57 ± 2.2% and the lowest in tumour 46 ± 1.3% (*p* < 0.00001). Moreover, values of ^15^N/^14^N expressed in delta notation were the highest in healthy tissues 9.84 ± 0.61 and the lowest in tumour 8.92 ± 0.58. Values of ^13^C/^12^C tended to be higher in tumour −22.2 ± 0.89 and the lowest in healthy tissues −23.7 ± 1.2. Tumour tissues differ in isotopic composition from tissues taken from margin and healthy tissues taken from distant oral mucosa.

## 1. Introduction

Worldwide data show that lip, oral cavity and pharyngeal cancer is one of the most noteworthy problems among oral diseases due to its significant effects on the quality of life of the patients. In 2017, among all cases of this type of cancer, more than half 57.4% are concerned lip and oral cavity cancers [1]. The global database showed that the complete number of incidences of lip and oral cavity cancers increased from 1990 to 2017 around 109% over this period [1]. In 2018, the number of new cases of lip and oral cavity cancer was 354,864 and the number ofdeaths was 177,384 [2].

The worldwide database showed that 630,000 new patients with head and neck cancer are diagnosed per year and almost 350,000 patients died. In total, 90% of head and neck cancers are squamous cell carcinoma raised from the mucosal surface of the oral cavity [3]. Demographic variation of this type of cancer depends on habits of tobacco use and alcohol consumption but also HPV (Human Papillomavirus) infections. In Northern America and Europe, oral squamous cell carcinomas (OSCC) accounts for 5% to 10% of all new cancer cases [3]. There is a noticeable decrease in cancers caused by smoking and drinking while the number of HPV-dependent tumors is increasing [3].

In 2012 in Europe, 140,000 new cases of head and neck cancers and 63,500 new deaths were reported. It was noted that men have a four times higher risk having this type of cancer then women [4].

In the United States head and neck cancers constitutes 3% of all malignancies. A total of 60,000 new cases are reported every year with approximately 12,000 resulting deaths.

From 2002 to 2012, incidence of cancer in men decreased 0.29% per year and the incidence in women decreased 0.38% per year [5].

In Poland, about 5.5–6.2% of all malignant tumours are head and neck cancers, which is manifested by 5500 to 6000 of new cases per year [6]. Oral cancers constitute 1.8% of all malignant neoplasms [7]. Cancers of this area cause approximately 3% of cancer-related deaths among men and about 1% among women in Poland. [6] A 5-year survival rate amounts to 47.6% for men and 49.1% for women [6].

Five-year age-standardised relative survival rates in Europe in 1999–2007 have been calculated for 39% for oropharynx cancer, 43% for tongue cancer, 45% for oral cavity cancer and 49% for nasopharynx [4]. Except for patients with laryngeal cancer, survival was significantly better in women than men [4].

Additionally, in Europe in 2012, mortality rate caused by oral cavity and pharynx cancer was 34,200 per 100,000 among male and 9400 per 100,000 among female [8].

Generally, mortality rate in OSCC is more than 50% [9]. Intervention at the earliest possible stage of the disease gives patients the best chance of survival. Unfortunately, a lot of cases are still diagnosed too late.

The most common method of treatment mainly depends on the tumour’s stage and grade [10,11]. However, some data suggest that tumours which are of the same stage may considerably vary in terms of physiology, response to treatment or prognosis [12]. As a consequence, in some cases it is impossible to diagnose an aggressive phenotype of cancer at an early stage [13,14]. Therefore, the best possible analysis of cancers at certain stages of development will facilitate efficient therapy, prognosis or predictions made-to-measure for patients [12,15,16,17]. It is important to devise additional prognostic methods which will enable the quickest way of detecting the disease. It is particularly important when it comes to cancers of the head and neck, as their surgical treatment influences patients’ quality of life. To provide patients’ the best opportunity for treatment, a highly sensitive and specific screening method for quick diagnosis and prognosis is needed.

A method used for evaluating molecular content of samples is mass spectrometry. One of its forms is isotope ratio mass spectrometry (IRMS) [18]. Isotopes are atoms of the same element that differ in the number of neutrons in the nucleus, while maintaining the same number of protons. Stable isotopes are non-radioactive forms of atoms. Compounds containing different isotopes of the same element may have slightly different reaction rates due to their differences in mass. These rate differences can lead to isotopic fractionation in the environment as compounds undergo physical, chemical and biological processes, and are manifested by differences in the ratio of heavy to light isotopic content.

The IRMS method is based on the measurement of the ratio of a heavier stable isotope to a lighter one, allowing one to detect either the enrichment or depletion in an examined sample (an increase or reduction in the heavier isotope content). The variances in the isotopic ratios of elements are expressed as delta values (δ). Relative measurements are used, by alternating measurements of a sample and a standard within a single measurement cycle and referring the measured isotopic content of the sample to the isotopic content of the standard.

This method allows to determine the ratio of isotopic composition of elements, which reflects the occurrence of physical or chemical reactions and metabolic processes. Data are used to obtain information on biological material through analysis of organic and non-organic compounds [19]. The most commonly performed is analysis of the isotopic composition of hydrogen, oxygen, carbon, nitrogen and sulphur—the most common elements occurring in proteins. Nitrogen and carbon elements have been selected for examination of tumor tissues, because they play a crucial role in the emergence and sustenance of cell metabolism and life, during cell growth and division, which are very important to cancer biology. Additionally, the isotopic composition of these two elements can be routinely tested at a level of precision sufficient for diagnostic applications. Moreover, it was recently shown that natural ^13^C and ^15^N isotope abundance may vary between healthy and breast cancer biopsy tissues. The isotope mass content has been related to lipid metabolism, anaplerosis and urea cycle, and thus to three pathways known to be altered in malignant cells. These findings may suggest that isotope balance of ^13^C and ^15^N is a good method of metabolism evaluation, because it reflects modifications in C partitioning and N excretion altogether [20].

Scientific research supports the statement that it is possible to characterise the origin of damaged or healthy cells by means of isotope ratio mass spectrometry [21,22]. Currently, it is used for laboratory diagnostics, genetics, biotechnology or proteomics [23,24,25,26,27]. Results of research carried out on human cancerous and non-cancerous tissues are available, but they more commonly include samples of hair, saliva or breath than of cancerous tissue directly [28,29,30,31,32]. Assessment of cancerous infiltration could lead to precise information about the stage of the disease at a molecular level. Scientific studies on cancerous tissues confirm that those results may help estimate the risk group in which the patient is. When it comes to oncology, IRMS builds doctors’ hopes on the possibility of reliable staging of the disease and adjusting the therapy to the patient. This method has already been used for analysis of Wilms’ tumour or rhabdomyosarcoma and it has been proven to obtain additional information on cancers’ characteristics [33]. When it comes to cancers of the head and neck, it is a new area of research that we want to explore in this study.

It should be emphasized that this method needs to be thoroughly validated before clinical implementation is warranted.

The aim of the study was to evaluate isotope ratio of nitrogen and carbon in oral squamous cell carcinomas (OSCC) as compared to margin and healthy tissue.

## 2. Experimental Section

A total of 18 patients of the Maxillofacial Department treated surgically due to malignant neoplasms of oral cavity were included in the study. Among them were 7 women and 11 men, aged 46–77 years (mean age 68.2 ± 7.8 SD). Smoking was identified in 6 of 11 of men and 4 of 7 of women.

The study group consisted consecutive patients that fulfilled inclusion criteria.

Including criteria were:Primary tumor of oral cavity, located in areas, such as labial or buccal mucosa, the anterior two thirds of tongue, the floor of the mouth, gingiva, alveolus and palate;The result of the histopathological examination confirmed that it is a planoepithelial cancer;Loco-regional advancement of the tumor enabling radical surgical treatment (T1–T3, N0–N2, M0).

Excluding criteria were:Presence of distant metastases;Patients after previous radio and chemotherapy treatment of the head and neck region;Loco-regional advancement of the tumor, not allowing to perform radical surgery.

Additional epidemiological and clinical features are presented in Table 1.

### 2.1. Preparation of Samples

For study purposes, a total of 180 tissue samples were collected from patients with oral squamous cell carcinomas at different stages of tumour.

The study was approved by the Bioethics Committee (RNN/185/18/KE).

During the surgery consisting of removing the primary tumor, collected from each patient were: 4 tumour samples, 4 margin samples and 2 healthy oral mucosa samples (40 mm away from the border of the tumour) with dimensions of ca. 2 × 2 mm. Four of them (2 from tumour and 2 from margin) were fixed in formalin for later histopathological verification. All samples were routinely reviewed, confirmed and evaluated by an experienced specialist in pathomorphology. The following features were assessed: depth of cancer infiltration (mm), angioinvasion, local lymph nodes metastases, nodal capsule infiltration and neuroinvasion with conventional hematoxylin and eosin staining, it is impossible to distinguish healthy tissue from a margin without neoplastic infiltration. However, it was possible to differentiate tissue with neoplastic infiltration from tissue without infiltration, i.e., healthy tissue.

### 2.2. IRMS Procedure

A part of tissue samples, 108 out of 180 intended for isotope ratio mass spectrometry (IRMS) procedure were kept at −70 °C before performing further procedures. The obtained tissue sections were frozen at −70 °C for another 48 h and freeze dried (lyophilizer Christ Delta 1–24 LSC, GmbH, Osterode am Harz, Germany). Subsequently, lyophilization procedure was performed.

For the IRMS measurements, three samples sized 3 ± 1 mg were prepared from each material. On average, three samples were prepared from each tissue specimen. Each one was weighted into a tin capsule to which 1 mg of vanadium pentoxide was added as a sulphur oxidation catalyst. All were rolled up carefully.

IRMS procedure was performed with use of Sercon SL20–22 Continous Flow Isotope Ratio Mass Spectrometer connected with a Sercon SL elemental analyser for simultaneous carbon-nitrogen-sulphur (NCS) analysis. The primary reference standard used was thiobarbituric acid.

Measurements of ^15^N/^14^N nitrogen isotopic composition, ^13^C/^12^C carbon isotopic composition and carbon-to-nitrogen mass C/N ratio were made. Isotopic ratios were described by delta values. Delta is a value that characterizes the ratio of heavier to lighter isotopes in relation to international standards for nitrogen (atmospheric, Air) and carbon (Pee Dee Belemnite, PBD). The results were also described by minimum and maximum values, mean value, median and standard deviation. The achievable precision of ^15^N and ^13^C spectrometer measurements was ±0.1–0.2.

### 2.3. Statistical Analysis

One-way analysis of variance was used for the detection of differences in tumour, margin and healthy tissues. Due to the presence of non-normal data distributions, the Kruskal–Wallis test was applied. The difference is considered significant if *p* < 0.05.

A comparison between dependences of clinical, histopathological features and data obtained from spectrometry was performed. The normality of the distributions was assessed by the Shapiro–Wilk test. Pearson correlation analysis (in normal distributions) or Spearman’s correlation analysis (otherwise) was used. Spearman’s correlation analysis, Student’s t-test and the Mann–Whitney test were used to compare the data. Stargraphics Centurion XVI, StarPoint Technologies. INC., The Plains, VA, USA, was used to perform statistical analyses.

## 3. Results

During the study, the percentage content of nitrogen and carbon was obtained in all samples. The results are presented as minimum, maximum, mean, standard deviations and median values. Details of conducted assessments are presented in Table 2.

It is noticeable that the highest content of nitrogen was observed in tumour tissues, medium in margin and the lowest in healthy tissues. The average nitrogen concentration in tumour tissue was 12 ± 0.4%, in margin 10 ± 0.8% and 8 ± 0.9% in healthy tissue. Average content of nitrogen turned out to be statistically significant. *p*-values of all of these observations were less than 0.00001.

When it comes to the percentage of carbon content, the results were exactly the opposite. The highest content was observed in healthy tissue, medium in margin and the lowest in tumour tissue. The average values were, respectively, 57 ± 2.2%; 51 ± 3.1% and 46 ± 1.3%. This content turned out to be statistically significant (*p* < 0.00001).

Assessment of total carbon-to-nitrogen ratio and nitrogen-to-carbon ratio was also performed, showing noticeable differences in these evaluations. Total carbon-to-nitrogen ratio was highest in healthy tissues 6.91 ± 1.12, then in margin 5.07 ± 0.64 and the lowest in tumour tissues 3.7 ± 0.24. The inverse was true for the nitrogen-to-carbon ratio. For healthy tissue, it was 0.15 ± 0.03, for margin 0.20 ± 0.03, and for tumour tissue 0.27 ± 0.02. A statistically significant difference amongst those values was observed. *p*-values were less than 0.00001. Details of performed estimations are presented in Table 2.

Finally, assessments of the isotopic composition of nitrogen and carbon were conducted.

Isotopic ratios were presented as delta values which are ratios of heavier to lighter isotopes in relation to international standards for nitrogen (atmospheric, Air—delta Air) and carbon (Pee Dee Belemnite, PBD—delta PDB).

It turned out that values of delta Air were the highest in healthy tissues, ranging from 8.35 (minimum) to 10.78 (maximum), with average 9.84 ± 0.61 and median 9.86, lower in margin tissues: 8.19 (minimum) to 11.38 (maximum), with average 9.53 ± 0.64 and median 9.49, and the lowest in tumour tissues 8.10 (minimum) to 10.88 (maximum), with average 8.93 ± 0.58 and median 8.71. The maximum, mean and median values were statistically significant when results of healthy tissue-tumour and margin-tumour were compared. *p* values were, respectively, *p* < 0.001, *p* < 0.0001, *p* < 0.001.

In addition, statistically significant differences in the minimum of delta Air were observed between healthy tissue-tumour, with *p*-values less than 0.01.

Values of delta PDB tended to be higher in tumour and ranged from −24.36 (minimum) to −19.99 (maximum), with average −22.21 ± 0.89 and median −22.33; in margin −24.93 (minimum) to −21.54 (maximum), with average −23.36 ± 0.92 and median −23.43, and in healthy tissues −25.26 (minimum) to −21.07 (maximum), with average −23.68 ± 1.18 and median −24.32.

The mean, standard deviation and median values were statistically significant when the results of healthy tissue-tumour and margin-tumour were compared. *p*-values were, respectively, *p* < 0.0001, *p* < 0.0001, and *p* < 0.001.

Besides *p*-value, maximum value between margin and tumour turned out to be at the level of statistical tendency.

Furthermore, a statistically significant difference of minimum value in all tissues was observed.

The isotopic picture of tumour, margin and healthy tissues is summarised in Table 3.

Comparison of clinical features with data obtained from spectroscopy revealed significant dependences in the case of body mass index (BMI) and alcohol consumption.

It turned out that correlations between BMI and nitrogen content in margin tissues and the mean value of delta PDB tumour were statistically significant. In the case of patients with higher BMI, the nitrogen content in margin tissues was also higher. The *p* value was less than 0.05.

Observations were inverse for values of delta PDB. The higher BMI index, the lower mean value of delta PDB tumour. The *p* value was less than 0.001.

Additionally, in patients who confirmed alcohol consumption, higher mean values of delta PDB tumour were noticed. The *p* value was less than 0.05.

Comparison of histopathological features demonstrated a correlation between the occurrence of angioinvasion and higher mean value of delta PDB tumour (*p* < 0.05).

Moreover, nodal capsule infiltration was associated with lower values of nitrogen content in tumour tissues (*p* < 0.05) and a lower N/C ratio in tumour tissues (*p* < 0.05). Details of conducted assessments are presented in Table 4, Table 5 and Table 6.

## 4. Discussion

To the best knowledge of this research team, until now there has been no research concerning the use of spectrometry for the assessment of head and neck cancers. It has never been used to evaluate oral squamous cell cancer tissues. This study is a pioneering attempt at filling this gap in science. The head and neck cancers’ occurrence keeps rising so it is essential to establish if spectrometry, this new possibility of evaluation, can be as helpful as other types of diagnosing malignancies. Some studies have already indicated usefulness of IRMS in case of Wilms’ tumour, hepatoblastoma or rhabdomyosarcoma [19,22,32]. Numerous articles prove that, based on neoplasm’s isotopic composition and ratios, it is possible to picture its abnormal metabolism that reflects the course of the disease [19]. A new method aimed at improving the process of staging may be a huge step forward in the assessment of the threat and towards winning our battle with the disease before it starts developing in an unstoppable manner

The strategy behind the study was to find the possible relation between composition of isotopes or the ratio between them, and histological and clinical characteristics of malignant tumours. These factors were compared in healthy human tissues, the cancer’s margin and neoplastic tissues. Through analysis of the data, it was established that each of the samples had distinctive parameters when it came to their origin, this way indicating the cancerous tissue. The highest average content of nitrogen was found in cancerous tissues. It has been demonstrated in previous studies that the phenomenon of isotope fractionation during various processes of synthesis, mostly deamination and transamination, may constitute an explanation [33,34,35,36,37]. Moreover, the ratio between isotopes may reflect the clinical advancement of the disease. The same observations were made by other scientists. Research confirmed that it was possible not only to identify the malignant tissue but also to establish the critical moment of dissemination of the disease and point out the cases of the most doubtful outcome [19].

The results of this study managed to depict the correlation between the isotopic content of different tissues and their origin—tumour, margin or healthy tissue. Various isotopic parameters that were measured could lead to the information on where the specimen was collected. The data seem sufficient to show the potential of the method to differentiate tumor from tumor margin and healthy tissue. The same phenomenon was observed by other researchers, who proved the possible use of IRMS for obtaining information on the biochemical processes and metabolism of affected cells which at this state showed abnormalities [38,39]. The discussed research also proved the existence of correlation between the isotopic composition of the cancer and its clinical characteristics. Results might allow for establishing a correlation between histopathological features (such as angioinvasion and nodule sac infiltration) and specific isotopic parameters. Moreover, the spectrometry method could make it possible to assess others risk factors, such as higher BMI index and alcohol consumption. This creates a great chance of using IRMS for predicting the course of the disease and assessing the development of the disease with one single examination. Other studies also proved that histopathologic factors such as metastatic spread or nodule infiltration could be detected in this type of analysis.

There are some important factors that may affect the final outcome when it comes to IRMS, but, if the procedure is carried out properly, they do not pose a real problem. For spectrometry, a tumour sample as small as 0.5 mg is enough for analysis; samples tested under this study were bigger. Moreover, the method of collecting the specimens is crucial—for example, the needle aspiration biopsy is not appropriate for this purpose; the correct method is cutting out pieces of tumour, as was done here. Samples collected for this study were properly preserved from decomposition and contamination, in accordance with regulations [37]. The team hopes to widen the study group in the future. The group of 18 patients provides a lot of information; however, the larger the study group, the more information can be obtained. Currently, also other novel promising techniques such as digital ex-vivo confocal microscopy are used to detect eventual positive margins in squamous cell carcinoma in fresh frozen tissue. This method is based on laser specimen scanning in two dimensions along the X and Y axis. Each sample is scanned twice with the use of a laser at wavelengths of 488 and 785 nm. The digital staining modality converts the fluorescence and reflectance into an image that is similar to convectional haematoxylin and eosin staining. Digital ex-vivo confocal imaging characterises with a high level of accuracy in margins assessment in nonmelanoma skin cancers. Moreover, it is a very fast method, requires minimal tissue preparation, gives a greater overview of the specimen compared to conventional frozen histopathology, and is not related with tissue loss [40]. IRMS described in this article is also a novel method used for the assessment cancer of the head and neck region. It requires a special tissue preparation—fresh frozen samples have to be lyophilizated for the further analysis. The advantage of this method is the fact that a small amount of collected material is sufficient for testing; moreover, the sampling procedure itself does not interfere with the routine diagnostic procedure. On the technical side, it does not require the participation of a specialist physician, expensive or complicated preparation procedures and valuable analysis is short and takes about 20 minutes [19]. The cost of materials used for sample testing is considerably low; however, it is higher compared to conventional histology. However, specialist expensive equipment is required [23]. Perhaps the solution of this problems may be the cooperation with suitably equipped units. This technique provides the information concerning the metabolic changes in tissue samples at the atomic level. In this study, it was observed that there are differences in isotope composition between: (1) healthy tissue and margin, (2) margin and cancer tissue, (3) healthy tissue and cancer tissue. In comparison, anatomopathological assessment enables us to distinguish pathological tissues from these without cancerous infiltration. Metabolomics has potential to be a reliable biomarker in all cancers that are associated in metabolic changes. Isotope ratio mass spectrometry is found to be the most versatile analytical technique worldwide with the potential for personalised medicine, to minimise the implementation of suboptimal regimes, minimise treatment failures and reduce treatment costs [19]. This method can have a clinical application in patient screening and tissue sample characterization, influencing prognosis in malignant tumors, related to alteration in C and N metabolism.

## 5. Conclusions

Isotope ratio mass spectrometry has proven itself to be a novel, very sensitive method of analysis with promising perspectives. It should be adapted, developed and used for personalised therapies. This study provided some relevant information on an area that has never been covered before—the use of isotope ratio mass spectrometry for analysis of head and neck cancers, providing an opportunity for a better-aimed treatment for patients with head and neck cancers thanks to specifying the risk and prognosis of their disease.

Tumour tissues differ in isotopic composition from tissues taken from margin and healthy tissues taken from distant oral mucosa. Additionally, a correlation between the content of isotope composition in oral cancer tissue compared with clinical and histopathological features was revealed.

## Figures and Tables

**Table 1 jcm-09-03760-t001:** Epidemiological data.

		Male	Female
Total Number of Patients		11	7
BMI (mean)		25.5 ± 4.1 SD	23.8 ± 4.9 SD
Age (mean)		66.7 ± 9.3 SD	70.4 ± 4.1 SD
Smoking intensity (number of cigarette packs per day × years of smoking)	<10	0	0
	10–20	0	2
	>20	6	2
Alcohol consumption (number of patients)		1	1
Previous metachronic cancer (number of patients)		2	1
Malignancy in family (number of patients)		2	1
Co-morbity (number of patients)	Cardiovascular diseases	8	3
	Metabolic diseases	3	1
	Others	4	5
TNM histopatological (number of patients)	T1	1	0
	T2	1	1
	T3	3	2
	T4	6	4
	N0	2	4
	N1	2	1
	N2	6	2
	N3	1	0
Grading (number of patients)	G1	0	0
	G2	8	5
	G3	3	2

**Table 2 jcm-09-03760-t002:** Percentage content of nitrogen and carbon in tumour, margin and healthy tissues and total carbon-to-nitrogen ratio C/N and nitrogen-to-carbon ratio N/C.

Percentage Content		Tumour	Margin	Healthy Tissue	*p* Value
Nitrogen	Min	11% (0.112)	9% (0.088)	6% (0.063)	
	Max	13 % (0.13)	12% (0.122)	10% (0.095)	
	Mean ± SD	12 ± 0.4% (0.124 ± 0.004)	10 ± 0.8% (0.102 ± 0.008)	8 ± 0.9% (0.083 ± 0.009)	<0.00001
	Median	13% (0.125)	10% (0.101)	8% (0.084)	
Carbon	Min	44% (0.44)	42% (0.42)	53% (0.53)	
	Max	51% (0.51)	57% (0.57)	61% (0.61)	
	Mean ± SD	46 ± 1.3% (0.46 ± 0.01)	51 ± 3.1% (0.51 ± 0.03)	57 ± 2.2% (0.57 ± 0.02)	<0.00001
	Median	46% (0.46)	51% (0.51)	57% (0.57)	
Total carbon-to-nitrogen ratio C/N	Min	3.43	3.40	5.60	
	Max	4.50	6.23	9.53	
	Mean ± SD	3.70 ± 0.24	5.07 ± 0.64	6.91 ± 1.12	
	Median	3.70	5.18	6.6	
Total nitrogen-to-carbon ratio N/C	Min	0.22	0.15	0.11	
	Max	0.29	0.29	0.23	
	Mean ± SD	0.27 ± 0.02	0.20 ± 0.03	0.15 ± 0.03	<0.00001
	Median	0.27	0.19	0.15	

**Table 3 jcm-09-03760-t003:** Results of ^15^N/^14^N and ^13^C/^12^C isotope ratio estimation.

		Delta Air	*p* Value *		Delta PDB	*p* Value **
Tumour	^15^N/^14^N Min	8.10	*p* < 0.01	^13^C/^12^C Min	−24.36	*p* < 0.00001
	^15^N/^14^N Max	10.88	*p* < 0.001	^13^C/^12^C Max	−19.99	*p* = 0.0514515
	^15^N/^14^N Mean +/− SD	8.93 ± 0.58	*p* < 0.0001	^13^C/^12^C Mean +/− SD	−22.21 ± 0.89	*p* < 0.0001
	^15^N/^14^N Median	8.71	*p* < 0.001	^13^C/^12^C Median	−22.33	*p* < 0.001
Margin	^15^N/^14^N Min	8.19		^13^C/^12^C Min	−24.93	*p* < 0.00001
	^15^N/^14^N Max	11.38	*p* < 0.001	^13^C/^12^C Max	−21.54	*p* = 0.0514515
	^15^N/^14^N Mean +/− SD	9.53 ± 0.64	*p* < 0.0001	^13^C/^12^C Mean +/− SD	−23.36± 0.92	*p* < 0.0001
	^15^N/^14^N Median	9.49	*p* < 0.001	^13^C/^12^C Median	−23.43	*p* < 0.001
Healthy Tissue	^15^N/^14^N Min	8.35	*p* < 0.01	^13^C/^12^C Min	−25.26	*p* < 0.00001
	^15^N/^14^N Max	10.78	*p* < 0.001	^13^C/^12^C Max	−21.07	
	^15^N/^14^N Mean +/− SD	9.84 ± 0.61	*p* < 0.0001	^13^C/^12^C Mean +/− SD	−23.68 ± 1.18	*p* < 0.0001
	^15^N/^14^N Median	9.86	<0.001	^13^C/^12^C Median	−24.32	*p* < 0.001

* The maximum, mean and median values differed statistically between: (1) healthy tissue and tumour (2) margin and tumour. In addition, statistically significant differences in the minimum of delta Air were observed between: (1) healthy tissue and tumour (*p* < 0.01). ** The comparison of mean, standard deviation and median values of healthy tissue and tumour as well as margin samples and tumour tissues revealed a statistical significance. The correlations between maximum values of margin and tumour samples were also observed (*p* value = 0.0514515). In addition, statistically significant differences in the minimum of delta PDB in all tissues were observed.

**Table 4 jcm-09-03760-t004:** Comparison of percentage content of nitrogen and carbon in tumour and margin and clinical features.

Clinical Features		*N*	Nitrogen Percentage Content in Tumour				Carbon Percentage Content in Tumour				Nitrogen Percentage Content in Margin			Carbon Percentage Content in Margin			
			Mean	Stat. Analysis Z	*t*	*p* Value	Mean	Stat. Analysis Z	*t*	*p* Value	Mean	*t*	*p* Value	Mean	Stati. Analysis Z	*t*	*p* Value
BMI		18	0.12		0.2856	0.7789	0.46		−1.3072	0.2096	0.1	2.2454	<0.05	0.51		−1.3072	0.2096
Depth of infiltration (mm)		18	0.12		0.3175	0.755	0.46		0.3807	0.7084	0.1	−0.0158	0.9876	0.51		0.8694	0.3975
Tabacco smoking	yes	10	0.13	−0.4		0.6965	0.46	0.04		0.9654	0.1	0.25	0.8039	0.5	−1.38		0.1728
	no	8	0.12				0.46				0.1	0.25	0.8039	0.52			
Alcohol consumption	yes	2	0.12	−0.91		0.3922	0.46	0.63		0.549	0.09	−2.59	0.1087	0.54	1.19		0.2614
	no	16	0.12	−0.91		0.3749	0.46	1.00		0.3283	0.1	−2.59	0.1087	0.51	−0.09		0.9298
Angioinvasion	yes	11	0.12	0,00		1.00	0.46	1.18		0.246	0.1	0.08	0.9359	0.51	0.39		0.7028
	no	7	0.13	−2.11		<0.05	0.46	1.36		0.1797	0.1	0.08	0.9359	0.52	0.61		0.5532
Local lumpoh nodes metastasis	yes	13	0.12	−1.08		0.2887	0.46	0.49		0.6331	0.1	−0.15	0.8856	0.51	0.69		0.5028
	no	5	0.12				0.45				0.1	−0.15	0.8856	0.51			
Nodal capsule infiltration	yes	6	0.12	−0.88		0.3865	0.47	1.77		0.077	0.1	−0.65	0.5263	0.52	1.32		0.1903
	no	12	0.13				0.46				0.1	−0.65	0.5263	0.51			
Neuroinvasion	yes	5	0.12	−0.47		0.6544	0.47	0.47		0.6544	0.1	−0.44	0.6688	0.52	1.66		0.1005
	no	13	0.13				0.46				0.1	−0.44	0.6688	0.51			

**Table 5 jcm-09-03760-t005:** Comparison of total nitrogen-to-carbon ratio N/C in tumour and margin and clinical features.

Clinical Features		*N*	[N]/[C] Tumour						[N]/[C] Margin				
			Mean	sd	Median	Statistical Analysis Z	*t*	*p* Value	Mean	sd	Median	*t*	*p* Value
BMI		18	0.27	0.02	0.27		1.0727	0.2993	0.2	0.03	0.19	1.4119	0.1771
Depth of infiltration (mm)		18	0.27	0.02	0.27		−0.4824	0.6361	0.2	0.03	0.19	0.0964	0.9244
Tabacco smoking	yes	10	0.27	0.00	0.27	−0.22		0.8286	0.2	0.04	0.2	0.19	0.8497
	no	8	0.27	0.02	0.28	−0.91		0.3922	0.2	0.03	0.19	−1.66	0.2978
Alcohol consumption	yes	2	0.27	0.00	0.27				0.17	0.03	0.17		
	no	16	0.27	0.02	0.27	−1.27		0.2109	0.2	0.03	0.2	0.67	0.5123
Angioinvasion	yes	11	0.27	0.02	0.27				0.2	0.04	0.19		
	no	7	0.28	0.01	0.28	−0.49		0.6331	0.19	0.02	0.2	0.01	0.9954
Local lumpoh nodes metastasis	yes	13	0.27	0.02	0.27				0.2	0.04	0.19		
	no	5	0.27	0.02	0.28	−2.2		<0.05	0.2	0.02	0.21	−0.75	0.4666
Nodal capsule infiltration	yes	6	0.26	0.02	0.27				0.19	0.03	0.19		
	no	12	0.28	0.01	0.28	−1.18		0.246	0.2	0.04	0.2	−0.48	0.6379
Neuroinvasion	yes	5	0.26	0.02	0.27				0.19	0.01	0.19		
	no	13	0.28	0.01	0.28	−1.77		0.077	0.2	0.04	0.2	−1.52	0.1479

**Table 6 jcm-09-03760-t006:** Comparison of ^15^N/^14^N and ^13^C/^12^C isotope ratio estimation in tumour and margin and clinical features.

Clinical Features		*N*	Delta Air Tumour Mean Value					Delta PDB Tumour Mean Value					
			Mean	sd	Median	*t*	*p* Value	Mean	sd	Median	Statistical Analysis Z	*t*	*p* Value
BMI		18	8.92	0.55	8.80	−0.2378	0.815	−22.21	0.78	−22.3		−4.3198	<0.001
Depth of infiltration (mm)		18	8.92	0.55	8.80	2.0287	0.0595	−22.21	0.78	−22.3		0.3049	0.7644
Tabacco smoking	yes	10	8.91	0.67	8.69	−0.12	0.9090	−21.93	0.81	−22.3	1.02		0.3154
	no	8	8.94	0.39	8.95	−1.48	0.2828	−22.56	0.64	−22.33	2.04		<0.05
Alcohol consumption	yes	2	8.57	0.32	8.57			−20.71	0.9	−20.71			
	no	16	8.97	0.56	8.83	0.99	0.336	−22.4	0.55	−22.33	2.26		<0.05
Angioinvasion	yes	11	9.03	0.65	8.85			−22,00	0.94	−22.08			
	no	7	8.76	0.32	8.65	1.04	0.3123	−22.53	0.28	−22.42	1.77		0.0754
Local lumpoh nodes metastasis	yes	13	9.01	0.61	8.85	1.36	0.1918	−22.07	0.87	−22.24	1.08		0.2908
	no	5	8.71	0.28	8.75			−22.56	0.38	−22.52			
Nodal capsule infiltration	yes	6	9.17	0.73	8.98	0.72	0.4800	−21.88	1.3	−21.82	0.59		0.5663
	no	12	8.80	0.42	8.78			−22.37	0.3	−22.33			
Neuroinvasion	yes	5	9.08	0.72	9.04			−22.2	1.07	−22.08			
	no	13	8.87	0.50	8.80			−22.21	0.7	−22.3			

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
