# Peer review of "Characteristic of Oral Squamous Cell Carcinoma Tissues Using Isotope Ratio Mass Spectrometry"

_jcm, 2020, doi:10.3390/jcm9113760_

Round 1

Reviewer 1 Report

The introduction needs to be more clear and include everything  in the methods to prepare the reader such as

what is isotope in details

explain more why nitrogen and carbon used in the analysis?

why did you include BMI without introducing that ?

LINE 247

NO research concerning  the use of spectrometry for assessment of head and neck cancers. How about OSCC? is this novel for OSCC?

Author Response

Point 1: What is isotope in details?

Response 1: A detailed and clarifying description is now included in the text (Lines 83-89)

Point 2: Explain more why nitrogen and carbon used in the analysis?

Response 2: An explanation has been included in the text: Nitrogen and carbon elements have been selected for examination of tumor tissues, because they play crucial role in the emergence and sustenance of cell metabolism and life, during cell growth and division, which are very important to cancer biology (Lines 100-109)

Point 3: Why did you include BMI without introducing that ?

Response 3: Thank you very much for this comment. This minor overlooked  had already been fixed. BMI was introduced in table 1. (Line 147)

Point 4: LINE 247 NO research concerning  the use of spectrometry for assessment of head and neck cancers. How about OSCC? is this novel for OSCC?

Response 4: According to the authors’ knowledge, it has never been used to evaluate directly to oral squamous cell carcinoma tissues.

We hope that after small changes introduction is now more clear and include everything what is needed

Reviewer 2 Report

This is an interesting report. The authors found that tumour tissues differ in isotopic composition from tissues taken from margin and healthy tissues taken from distant oral mucosa.Minor changes are needed:

  1. In the discussion please add some information about the current use of promising new tecniques to detect eventual positive margins in squamous cell carcinoma, as ex-vivo confocal microscopy and compare with a speculation the two tecniques. In this regard add in the reference the article "Digital ex-vivo confocal imaging for fast Mohs surgery in nonmelanoma skin cancers: An emerging technique in dermatologic surgery. Dermatol Ther. 2019 Nov;32(6):e13127. doi: 10.1111/dth.13127. Epub 2019 Nov 12. PMID: 31628777." 

      2.  In the discussion please add also the the possible feasibility of using this   technique during daily clinical practice and the related costs

3. Please add some information about the time of this tecnique. Is it a save time tecniques?

Author Response

Point 1: In the discussion please add some information about the current use of promising new techniques to detect eventual positive margins in squamous cell carcinoma, as ex-vivo confocal microscopy and compare with a speculation the two techniques.

Response 1: Additional data about the new innovative research method and its comparison to the IRMS technique was made and was presented in the selected lines of the text. (Lines 315-342)

Point 2: In this regard add in the reference the article "Digital ex-vivo confocal imaging for fast Mohs surgery in nonmelanoma skin cancers: An emerging technique in dermatologic surgery. Dermatol Ther. 2019 Nov;32(6):e13127. doi: 10.1111/dth.13127. Epub 2019 Nov 12. PMID: 31628777."

Response 2: Article was added to the references. Reference number 40. (Lines 484-487)

Point 3: In the discussion please add also the possible feasibility of using this technique during daily clinical practice and the related costs.

Response 3: All the collected information on this problem were added in the text (Lines 315-342)

Point 4: Please add some information about the time of this technique. Is it a save time techniques?

Response 4: On the technical side does not require the participation of a specialist physician, expensive or complicated preparation procedures and valuable analysis is short and takes about 20 minutes. Lines (315-342)
